# Drug overdose among women in intimate relationships: The role of partner violence, adversity and relationship dependencies

Nabila El-Bassel[1], Phillip L. Marotta[2]*, Dawn Goddard-Eckrich[1], Mingway Chang[1], Tim Hunt[1], Ewin Wu[1], Louisa Gilbert[1]

1 Columbia University, School of Social Work, New York, New York, United States of America, 2 Yale University, School of Medicine, Department of Psychiatry, New Haven, Connecticut, United States of America

* phillip.marotta@yale.edu

**Data Availability Statement:** The data underlying the results of the study have been uploaded as Supporting Information. Additional data are

## Abstract

### Background

This study examines the relationship between experiencing intimate partner violence (IPV), exposure to prior childhood adversity, lifetime adverse experiences, drug-related relationship dependencies with intimate partners and overdose, hospitalization for drug use, friends and family members who overdosed and witnessing overdose.

### Methodology

This paper included a sample of 201 women who use drugs in heterosexual relationships with criminal justice-involved men in New York City. We included measures of experiencing overdose, hospitalization for drug use, witnessing overdose, and having friends and family who overdosed. Intimate partner violence consisted of either 1) none/verbal only, 2) moderate and 3) severe abuse. Dichotomous indicators of drug-related relationship dependencies included financial support, drug procurement, splitting and pooling drugs. A scale measured cumulative exposure to childhood adversity and lifetime exposures to adverse events. This paper hypothesized that experiencing moderate and severe IPV, drug-related dependencies and exposure to prior childhood and lifetime adversity would be associated with a greater risk of experiencing overdose, hospitalization for drug use, witnessing overdose and having friends and family members who overdosed. Generalized linear modeling with robust variance estimated relative risk ratios that accounted for potential bias in confidence intervals and adjusted for race, ethnicity, education and marital status.

### Results

We found experiencing moderate or severe IPV was associated with ever being hospitalized for drug use and having a family member who experienced overdose. Experiencing moderate IPV was associated with increased risk of witnessing overdose, Partner drug dependencies were associated with overdose, ever being hospitalized for drug use, witnessing overdose, and having a family member or friend who experienced overdose. Childhood and

available via Columbia Academic Commons using the following DOI: doi.org/10.7916/d8-rm2n-hv16.

**Funding:** National Institute on Drug Abuse R01 DA033168 to NE, National Institute on Drug Abuse T32 DA019426 to PM, and National Institute on Drug Abuse F31 DA044794 to PM supported this work. The funders had no role in study design, data collection and analysis, decision to publish, or preparation of the manuscript.

**Competing interests:** The authors have declared that no competing interests exist.

lifetime adversity exposures were significantly associated with increased risk of overdose, ever being hospitalized for drug use, ever witnessing overdose and having a friend and family member who overdosed.

## Conclusion

Findings underscore the intersection of experiencing IPV and drug-related relationship dependencies, childhood adversity and lifetime adversity in shaping experiences of and witnessing overdose among women who use drugs. They highlight the urgent need to address IPV, adversity experiences and drug-related relationship dependencies in overdose prevention for women who use drugs.

## Introduction

In the United States, drug overdose rates among women have increased exponentially over the past two decades.[1][2][3] Overdose rates among women between 30–65 in the United States increased by 260% from 1999–2017 with rates growing the most due to overdose from opioids. [3] Between 1999–2015, opioid overdose deaths increased by 471% among women compared to 218% among men.[4][5] The greatest increase occurred in deaths involving synthetic opioids, which increased by 850% among women during the same period.[4][5] Racial and ethnic disparities exist in the rates of overdose, in that black women are disproportionately impacted by overdose from opioids particularly fentanyl.[6] Between 2011–2016, overdose deaths from fentanyl increased the most among African Americans with a growth of 140.6% per year compared to Hispanics (118.3%) and whites (108.8%).[6] Despite rapidly rising rates of fatal and non-fatal overdose, research on risk factors associated with overdose specifically among women is limited.

### Intimate partner violence, relationship dependencies and overdose

The risk environment framework provides a multi-level conceptualization of mechanisms that explain the association between overdose and experiencing IPV, a history of child adversity and adverse life experiences.[7][8][9][10] The risk environment perspective presumes that intrapersonal factors including intimate partner violence, gender norms and relationship power inequities interacts with individual characteristics to shape the occurrence of drug use, and risk of drug-related harms of overdose.[8] A risk environment that includes being forced to have sex against one's will or without a condom as well as being kicked, slammed against a wall, beaten, punched, and choked is associated with increased substance use dependence, greater frequency/quantity of use among women and also may lead to increased risk of overdose.[11][12][13][14][15][16]

Prior research has shown that following hospitalization for drug use, people who use opioids are at greater risk of returning to drug use and overdose due to lower tolerance and experiencing symptoms of withdrawal.[17] Emergency department hospitalization is an important yet understudied feature of the overdose risk environment for women who use opioids. Hospitals may be an opportune setting of the risk environment to deliver both overdose prevention through naloxone distribution and IPV services for women who are hospitalized due to drug use.[17] No studies to date have examined the associations among women between IPV and the specific outcomes of overdose, hospitalization from drug use, witnessing overdose, and having friends or family who have overdosed.

One explanatory mechanism for relationships between IPV and overdose is that women may use drugs to cope with negative affective experiences due to prior exposure to adversity such as IPV and childhood abuse (Carbone-Lopez et al., 2006; El-Bassel et al., 2004; El-Bassel et al., 2014; Lipsky et al., 2005).[13][18][19] Prior research by El-Bassel et al[12] and Amaro et al[20] found that women who experienced IPV coped with adverse experiences by using drugs, which could lead to overdose. Testa et al[16] found that use of illicit drugs was associated with an increased risk of IPV compared to women who did not use illicit drugs. Conversely, women who use drugs may be at greater risk of experiencing IPV because partners may perceive them as vulnerable to victimization, an increased risk of relationship conflict, lack of resources to leave violent partners and other factors.[16][21][22][23]

Inequities in drug-related relationship dependencies is another factor of the micro risk environment that may increase the possibility of overdose for women.[23][24][25] Women are more likely to rely on intimate partners to procure illicit substances and to teach them how to administer drugs; thus, shaping the unique microsocial contexts of overdose risk environments.[19][25] New insights regarding relationship dependencies and overdose could inform future research into couple-focused interventions that incorporate both partners' drug use and sharing behaviors into overdose prevention interventions for women who use drugs.

**Childhood and lifetime adverse exposures and overdose.** In addition to IPV, literature suggests that coping with prior adverse experiences experienced during childhood and throughout women's lifetime is an individual-level factor of the risk environment that may heighten risk of overdose.[26][27][28] Childhood adverse experiences may create a vulnerability to using greater quantities and higher frequencies of substance use to cope with the psychological sequelae of trauma; thus, increasing the risk of overdose.[26][27][28] Within the risk environment framework, childhood adversity and exposure to violence in addition to IPV across the life-course are individual factors that predispose women to greater vulnerabilities to using substances and experiencing overdose later in life.[28] Lake et al.[27] found that exposure to physical, sexual, emotional and physical abuse and neglect were associated with non-fatal overdose in a sample of 552 women.

**Markers of overdose risk among women who use drugs.** Witnessing overdose or having friends and family members who have experienced overdose is a factor of the overdose risk environment that emphasizes the importance of overdose prevention within the micro-social contexts of interpersonal relationships.[29][30] People who use drugs are embedded in familial and social networks with many other drug-using individuals; thus, increasing their risk of knowing others who have overdosed. Research shows that people who use drugs are at greater risk of witnessing overdose compared to people who do not use such substances.[29]30[31] It is estimated that from 58% to 86% of all overdoses occur in the presence of witnesses who could potentially intervene during the observed overdose.[32] A systematic review by Martins et al[33] found lifetime prevalence of witnessing a drug overdose was 73.3% with a range of 50% to 96% in 17 studies of people who use drugs. Having family members and friends who experienced overdose is an understudied social factor of the overdose risk environment.

## Gaps in the literature

Several gaps persist in research examining the overdose risk environment of women who use drugs in intimate partnerships. First, women, particularly racial and ethnic minorities, are underrepresented in extant research on factors of the risk environment that are associated with overdose in the United States despite mounting literature suggesting exposure to adversities are associated with greater severity of substance use. Second, research on overdose prevention neglects risk factors that disproportionately impact women, namely IPV and childhood

adversity. Third, the association between drug-related relationship inequities and overdose among women remains unknown. Finally, little is known about how exposure to IPV and drug-related relationship inequities shape social factors of the overdose risk environment including witnessing overdose and having friends and family members who have overdosed. Investigating the association between IPV and overdose risk factors could inform future overdose prevention interventions to address the unique risk factors facing women who use drugs in the United States.

The following paper described the frequency of overdose, witnessing overdose of friends and family members, and hospitalization due to drug use in a sample of women in New York City. We examined the association between the experience of IPV and several indicators of overdose risk that included 1) experiencing overdose, 2) being hospitalized because of drug use, 3) witnessing overdose, and 4) having family and 5) friends who experienced overdose after adjusting for covariates of race, ethnicity, less than high school education and marital status. This paper also examined the association between drug-related relationship inequities (financial, splitting drugs, pooling drugs) and indicators of overdose risk after adjusting for potential confounders. Finally, this paper investigated the association between exposure to childhood and other lifetime adversities and overdose risks after adjusting for potential confounders. We hypothesized that 1) experiencing intimate partner violence and 2) drug-related relationship dependencies as well as lifetime exposure to 3) childhood adversity and 4) lifetime adversity would be associated with indicators of overdose risk after adjusting for potential confounders.

## Methods

### Data and procedures

Data consisted of a subset of 201 women who participated in Project PACT, a couple-focused randomized clinical trial of an HIV prevention intervention for men undergoing community corrections and their female intimate partners.[34] Following sentencing, participants were recruited from community correction provider sites by research assistants who provided informational fliers to male clients.[34] Because data was from a couples-focused intervention, women were recruited through first screening their male partners. All male partners were involved in community corrections at the time of recruitment and identified their female partners for inclusion in the study. Women who consented were administered a screening instrument that determined eligibility. The sample was restricted to include only women who reported lifetime use of illicit drugs from a starting sample of 239 women who were partners of men in community corrections. The Columbia University Institutional Review Board approved this study protocol and written consent was obtained from all participants.

**Inclusion/Exclusion criteria.** Couples were eligible to participate in the study if 1) they were at least 18 years of age, 2) identified each other as their primary partner, 3) the length of the relationship was greater than 3 months 4) they had unprotected vaginal or anal sex within the past 90 days 5) at least one partner had exposure to an outside HIV risk in the past year including unprotected sex with another partner, shared syringes, tested positive for an STI or HIV, 6) the male partner had the male partner had been mandated to some form of community supervision in the past 90 days (e.g. probation, parole, ATI, drug court, community court).

### Measures

Participants were interviewed by trained research staff and administered a structured questionnaire. All participants were reimbursed 265$ for participation in all research-related

activities including baseline questionnaire including screening, baseline, biological specimen and follow up assessments.

**Dependent variables.** Variables related to overdose included 1) experiencing overdose, 2) hospitalization due to drug use, 3) witnessing an overdose, and 3) having friends and family members who had experienced overdose. Overdose consisted of a dichotomous variable based on self-reported lifetime and past year experience of overdose (losing consciousness) while using drugs and whether the drugs were heroin, opiate pain relievers or tranquilizers. Hospitalization because of drug use included having ever received emergency room treatment for drug or alcohol use-related problems (overdose, loss of consciousness, physical problems or injuries caused by use of drugs or alcohol). Witnessing an overdose included a dichotomous variable indicating ever witnessing an overdose (loss of consciousness) while using drugs. Dichotomous variables measured having one or more 1) family members or 2) friends experience an overdose from using drugs.

**Independent variables.** Intimate partner violence consisted of a categorical variable with 21 questions measuring intimate partner violence based on the Conflict Tactics Scale with high reliability and validity.[35] A 4-level categorical variable was created with categories measuring exposure to 0) no physical/sexual, 1) verbal aggression only (no moderate /severe physical or sexual abuse) 2) only moderate and 3) severe sexual/physical abuse.

Lifetime adverse experiences was based on the Stressful Life Events Screening Questionnaire.[35] The scale summed question items consisting of dichotomous variables of ever experiencing 1) physical assault, 2) being threatened with a weapon, 3) witnessing death/injury, 4) being in an extremely frightening situation, 5) having a loved one who died because of accident, suicide or homicide, 6) experiencing physical force in a robbery and 7) experiencing serious injury and 8) other situation where life was in danger (Chronbach's alpha .87) (range 0–8).

Childhood adverse experiences consisted of a scale based on the Revised Inventory of Adverse Childhood Experiences summing 6 dichotomous variables.[36] (Questions measured 1) verbal *('Before the age of 18, has a parent or caregiver repeatedly ridiculed you, put you down, ignored you, or told you were no good?')* 2) sexual *(Before the age of 17, did anyone ever touch private parts of your body, made you touch their body, or made you have sex against your wishes?)*, and 3) physical abuse, *('Before the age of 17, did a parent, caregiver or other person ever slap you repeatedly, beat you, or otherwise attack or harm you?')*, 4) witnessing intimate partner violence *('Before the age of 17, did you ever witness your parents hitting, slapping, beating or physically hurting each other?')*, 5) having been placed in foster *care ('Before the age of 17, were you ever removed from your home and placed in foster care?')* and 6) running away from home *('Before the age of 17, did you run away from home?')* (Chronbach's alpha = .89) (range 0–6).[37]

Drug-related relationship dependencies included dichotomous variables measuring if more than half the time participants 1) purchased drugs for intimate partners, 2) relied on intimate partners to purchase their drugs, and 3) pooled and 4) split drugs with their intimate partner. Question items were summed to produce a scale reflecting cumulative dependencies on partners for drugs (Chronbach's alpha = .84) (range 0–4).

Socioeconomic factors consisted of a categorical variable measuring race (African American, Non-Hispanic White, Asian), dichotomous indicators of Hispanic ethnicity, less than high school education and marital status, and a continuous variable measuring age.

Drug and alcohol use included dichotomous variables indicating lifetime use of binge drinking (4 or more drinks within 5 hours) and use of heroin, prescription pain relievers, cocaine, crack, stimulants, tranquilizers and other drugs.

## Statistical analyses

**Descriptive analyses.** Descriptive statistics provided proportions and counts for dichotomous and categorical data as well as median and interquartile range (IQR) estimates for continuous variables. Bivariate analyses consisted of tests for significant differences between adversity exposure variables and experiencing and witnessing overdose as well as having friends and family members who experienced an overdose using chi-square tests comparing categorical variables and Wilcoxon Rank Sum tests for significant differences between categorical and continuous variables.[38][39][40]

**Hypothesis testing in unadjusted and adjusted models.** Generalized linear modeling tested the hypotheses that intimate partner violence, relationship dependencies, childhood and lifetime adversity would be associated with 5 indicators of overdose risk (experiencing overdose, hospitalization due to drug use, witnessing an overdose, having friends and family who experienced overdose).[41][42][43][44][45] Hypothesis tests of IPV, childhood sexual abuse, lifetime experiences of adversity and drug-related relationship dependencies were performed in separate models to avoid potential issues of collinearity.[39][42]

Parameter estimates of relative risk ratios (RR) tested hypotheses by measuring the likelihood of overdose risk and other overdose indicators based on exposure to intimate partner violence, drug-related relationship dependencies, childhood and lifetime adverse compared to individuals who were not exposed to adverse experiences.[43][45] Sensitivity analysis with other variables were performed with age, binge drinking and several types substances (i.e cocaine, crack, heroin). None of these variables were significant and due to the small sample size were not included in the final regression model estimating the association between IPV and overdose. Final models adjusted for race, ethnicity, education, and marital status. All participants reported lifetime illicit drug use prohibiting covariance adjustment for illicit drug use. The error variance correction method provided estimates that were robust to bias in confidence intervals.[46] All analyses were performed in STATA 15.[47]

## Results

### Descriptive findings

**Indicators of overdose risk.** Overall 12.4% of the women reported experiencing overdose in their lifetime (n = 25) (Table 1). Of those who experienced overdose, 6.0% (n = 12) reported overdose on heroin, 5.0% (n = 10) overdosed on benzodiazepines and 3.0% (n = 6) overdosed on opiates. Hospitalization due to drug or alcohol use was reported by 16.9% (n = 34) of the sample. More than a quarter reported having a friend (29.3%, n = 59) and a fifth reported a family member (20.9%, n = 42) who overdosed. More than a quarter of the sample reported witnessing an overdose at some point in their life (27.4%, n = 55).

**Intimate partner violence.** More than a fifth of the sample reported history of experiencing severe (21.4%, n = 43), nearly a fifth reported moderate abuse (17.4%, n = 35) and 11.0% (n = 22) reported only verbal abuse by current intimate partners.

**Lifetime and childhood adversity**. The median exposures to adversity was 2.1 for childhood adversity (IQR = 0, 6.0) and 2.3 (IQR = 0,8) for lifetime adversity.

**Relationship dependencies**. The most prevalent relationship dependency was providing money to partners to purchase drugs (33.3%, n = 67) followed by depending on partners to buy them drugs (30.9%, n = 62), pooling drugs with partners (11.0%, n = 22) and splitting drugs with partners (16.9%, n = 34). The mean score for relationship dependencies was 2.00 (0–4).

**Table 1. Descriptive characteristics of women who ever used illicit drugs (n = 201).**

| | | | Female partners | |
|---|---|---|---|---|
| | | | % or median | (n or IQR) |
| **Overdose** | | | | |
| | Lifetime | | 12.4 | (25) |
| | Type of drug | | | |
| | | Heroin | 6.0 | (12) |
| | | Other opiates | 3.0 | (6) |
| | | Benzodiazepine | 5.0 | (10) |
| Ever hospitalized from drug/alcohol use | | | 16.9 | (34) |
| Social networks | | | | |
| | Friend overdose | | 29.3 | (59) |
| | Family member overdose | | 20.9 | (42) |
| Witnessing overdose | | | | |
| | Ever witness overdose | | 27.4 | (55) |
| Access to naloxone | | | | |
| | | Ever heard of naloxone | 13.4 | (27) |
| | | Ever talked with others about naloxone | 8.7 | (18) |
| | | Ever used naloxone to reverse overdose | 2.0 | (4) |
| **Interpersonal factors** | | | | |
| | Intimate partner violence IPV) | | | |
| | | None | 50.3 | (101) |
| | | Verbal only | 11.0 | (22) |
| | | Moderate sexual/physical abuse only | 17.4 | (35) |
| | | Severe sexual/physical abuse | 21.4 | (43) |
| Relationship dependencies | | | | |
| | | Drug dependency | 30.9 | (62) |
| | | Drug provider | 33.3 | (67) |
| | | Pool money to buy drugs | 11.0 | (22) |
| | | Do drugs or split drugs | 16.9 | (34) |
| | Childhood adversity | | 2.1 | (0, 6) |
| | Lifetime adversity | | 2.3 | (0, 8) |
| Illicit drug use | | | | |
| | | Binge drinking | 46.8 | (94) |
| | | Heroin | 24.4 | (49) |
| | | Prescription pain relievers | 19.4 | (39) |
| | | Cocaine | 40.3 | (81) |
| | | Crack | 29.4 | (59) |
| | | Stimulants | 8.5 | (17) |
| | | Tranquilizers | 15.4 | (31) |
| | | Other drugs | 30.9 | (62) |
| Black race | | | 70.7 | (142) |
| Hispanic ethnicity | | | 30.4 | (61) |
| Income (<850$/month) | | | 53.2 | (107) |
| Education | | | 33.3 | (67) |
| Married | | | 31.8 | (64) |
| Age | | | 34.8 | (23.09, 64.85) |

**Overdose and hospitalization.**

**Socioeconomic factors**. A majority of the sample were black (70.7%, n = 142) and 30.4% (n = 61) were of Hispanic ethnicity. A third of the sample reported less than a high school education (33.3%, n = 67) and 31.8 (n = 64) were married. The mean age of the sample was 34.8 years (IQR = 23.09, 64.85).

*Drug use.* Nearly a quarter of women reported using heroin (24.4%, n = 49) in their lifetime and nearly a fifth reported using prescription pain relievers (19.4%, n = 39). Tranquilizer use was reported by 15.4% (n = 31) of the sample. Cocaine use was reported by 40.3% (n = 81) and crack cocaine was reported by 29.4% (n = 59) of the women.

**Hypothesis 1: Intimate partner violence and indicators of overdose risk.** Multivariable analysis did not rule out the null hypothesis regarding an association between severe IPV (3) or moderate IPV (2) compared to those reporting no physical/sexual IPV (0) and experiencing overdose (Table 2). Experiencing moderate ($RR_{adjusted}$ = 2.1, 95% CI = 1.0, 4.6) and severe IPV ($RR_{adjusted}$ = 2.0, 95% CI = .9, 4.2) were associated with increased risk of reporting prior hospitalization for drug use. Experiencing moderate IPV was significantly associated with increased risk of ever witnessing an overdose compared to participants who were not exposed to IPV ($RR_{adjusted}$ = 1.9, 95% CI = 1.1, 3.2) (Table 3). Exposure to moderate IPV ($RR_{adjusted}$ = 2.0, 95% CI = .9, 4.1) and severe IPV ($RR_{adjusted}$ = 2.1, 95% CI = .9, 4.1) was associated with increased risk of having a family member who experienced an overdose (Table 4).

**Hypothesis 2: Childhood adverse events and indicators of overdose risk.** Each additional exposure to childhood adverse events was associated with an increase in the risk of experiencing overdose ($RR_{adjusted}$ = 1.3 95% CI = 1.1, 1.6). Exposure to childhood adversity was associated with an increase in risk of prior hospitalization due to drug use ($RR_{adjusted}$ =

**Table 2. Bivariate and multivariate associations between exposures to intimate partner violence, adversities, relationship factors, and overdose and hospitalization for women who reported ever using illicit drugs (n = 201).**

| | | Overdose ever | | | | Ever hospitalized for drug use | | | |
| --- | --- | --- | --- | --- | --- | --- | --- | --- | --- |
| | | Unadjusted | | Adjusted | | Unadjusted | | | Adjusted |
| | | RR | 95%CI | RR | 95%CI | RR | 95%CI | RR | 95%CI |
| *Interpersonal trauma* | | | | | | | | | |
| Intimate partner violence | | | | | | | | | |
| | None | ref | ref | ref | ref | ref | ref | | ref |
| | Verbal | .8 | (.2, 3.5) | - | - | 1.1 | (.4, 3.8) | 1.2 | (.4, 3.9) |
| | Moderate | 1.3 | (.5, 3.5) | - | - | 2.2 | (1.0, 4.7) | 2.1 | (1.0, 4.6) |
| | Severe | 1.5 | (.6, 3.6) | - | - | 2.0 | (1.0, 4.2) | 2.0 | (.9, 4.2) |
| Relationship dependencies | | | | | | | | | |
| | Drug financial dependency | 2.4 | (1.2 5.0) | 2.0 | (1.0, 4.0) | 1.8 | (1.0, 3.3) | 1.7 | (.9, 3.2) |
| | Drug financial provider | 3.6 | (1.7, 7.6) | 2.9 | (1.3, 6.2) | 3.2 | (1.7, 6.1) | 3.1 | (1.7, 5.8) |
| | Pool money to buy drugs | 5.4 | (2.8, 10.6) | 4.4 | (2.0, 9.7) | 2.5 | (1.3, 4.8) | 2.4 | (1.2, 4.8) |
| | Do drugs or split drugs | 4.5 | (2.3, 9.1) | 3.9 | (1.9, 8.0) | 3.0 | (1.7, 5.5) | 3.0 | (1.7, 5.5) |
| | Scale | 1.7 | (1.3, 2.0) | 1.6 | (1.2, 2.0) | 1.4 | (1.2, 1.7) | 1.4 | (1.2, 1.7) |
| Childhood adversity | | 1.3 | (1.1, 1.6) | 1.3 | (1.1, 1.6) | 1.2 | (1.0, 1.4) | 1.2 | (1.0, 1.4) |
| Lifetime adversity | | 1.2 | (1.1, 1.4) | 1.2 | (1.0, 1.3) | 1.2 | (1.1, 1.3) | 1.2 | (1.1, 1.3) |

Adjusted models included covariates of race, ethnicity, education, marital status; RR: Relative Risk

**Table 3. Bivariate and multivariate associations between exposures to intimate partner violence, adversities, relationship dependencies and witnessing overdose for women who reported ever using illicit drugs (n = 201).**

| | | Ever witness overdose | | | |
| --- | --- | --- | --- | --- | --- |
| | | Unadjusted | | Adjusted | |
| | | RR | 95% C.I | RR | 95% C.I |
| *Interpersonal trauma* | | | | | |
| **Intimate partner violence** | | | | | |
| | None | ref | ref | ref | ref |
| | Verbal | 1.0 | (.5, 2.4) | 1.0 | (.4, 2.3) |
| | Moderate | 1.9 | (1.0, 3.0) | 1.9 | (1.1, 3.2) |
| | Severe | 1.3 | (.7, 2.3) | 1.2 | (.7, 2.2) |
| Relationship dependencies | | | | | |
| | Drug financial dependency | 1.5 | (1.0, 2.3) | 1.4 | (.9, 2.2) |
| | Drug financial provider | 1.5 | (1.0, 2.4) | 1.5 | (1.0, 2.4) |
| | Pool money to buy drugs | 2.8 | (1.8, 4.2) | 2.7 | (1.7, 4.2) |
| | Do drugs or split drugs | 2.0 | (1.3, 3.2) | 1.9 | (1.2, 3.0) |
| | Scale | 1.3 | (1.1, 1.5) | 1.2 | (1.1, 1.4) |
| Childhood adversity | | 1.3 | (1.1, 1.4) | 1.3 | (1.1, 1.4) |
| Lifetime adversity | | 1.3 | (1.2, 1.4) | 1.3 | (1.2, 1.4) |

Adjusted models included covariates of race, ethnicity, education, marital status; RR: Relative Risk

1.2, 95% CI = 1.0, 1.4). Each additional exposure to childhood adversity was associated with greater risk of having witnessed an overdose ($RR_{adjusted}$ = 1.3, 95% CI = 1.1, 1.4). Each additional exposure to childhood adversity ($RR_{adjusted}$ = 1.2, 95% CI = 1.1, 1.4) was associated with an increase in risk of a family member experiencing an overdose. Each additional exposure to childhood adversity was associated with greater risk of having a friend who overdosed ($RR_{adjusted}$ = 1.1, 95% CI = 1.0, 1.3).

**Hypothesis 3: Lifetime adverse experiences and indicators of overdose risk.** Each increase in lifetime exposure to adversity was associated with greater risk of experiencing an overdose ($RR_{adjusted}$ = 1.2, 95% CI = 1.0, 1.3). Each additional lifetime exposure to adversity was associated with greater risk of witnessing an overdose after adjusting for potential confounders ($RR_{adjusted}$ = 1.2, 95% CI = 1.1, 1.3). Each additional increase in lifetime exposures to adversity was associated with greater risk of a family member experiencing an overdose ($RR_{adjusted}$ = 1.2, 95% CI = 1.1, 1.3). Each additional exposure to lifetime adverse experiences was associated with greater risk of friends experiencing an overdose ($RR_{adjusted}$ = 1.2, 95% CI = 1.1, 1.3).

**Hypothesis 2: Relationship dependencies and indicators of overdose risk.** Participants who reported relying on their partners to buy them drugs ($RR_{adjusted}$ = 2.0, 95% CI = 1.0, 4.0), paying for their partners' drugs ($RR_{adjusted}$ = 2.9, 95% CI = 1.3, 6.2), pooling money to buy drugs ($RR_{adjusted}$ = 4.4, 95% CI = 2.0, 9.7), and doing drugs or splitting drugs with their partners ($RR_{adjusted}$ = 3.9, 95% CI = 1.9, 8.0) were associated with increased relative risk of ever experiencing an overdose. Each additional relationship dependency was associated with an increase of 60% in the risk of experiencing overdose ($RR_{adjusted}$ = 1.6, 95% CI = 1.2, 2.0). Participants who paid for their partners' drugs ($RR_{adjusted}$ = 3.1, 95% CI = 1.7, 5.8), pooled money to buy drugs ($RR_{adjusted}$ = 2.4, 95% CI = 1.2, 4.8) and did drugs or split drugs with their intimate partners ($RR_{adjusted}$ = 3.0, 95% CI = 1.7, 5.1) were more likely to report having a prior hospitalization due to substance use. Each additional relationship dependency was associated

**Table 4. Bivariate and multivariate associations between exposures to intimate partner violence, adversities, relationship dependencies and family and friends experiencing overdose for women who reported ever using illicit drugs (n = 201).**

| | | Friends experience overdose | | | | Family experience overdose | | | |
|---|---|---|---|---|---|---|---|---|---|
| | | Unadjusted | | Adjusted | | Unadjusted | | Adjusted | |
| | | RR | 95% C.I | RR | 95% C.I | RR | 95% C.I | RR | 95% C.I |
| **Intimate partner violence** | | | | | | | | | |
| | None | ref | ref | ref | ref | ref | ref | ref | ref |
| | Verbal | .8 | (.3, 1.8) | - | - | 2.0 | (.9, 4.6) | 1.9 | (.84, 4.4) |
| | Moderate | 1.1 | (.6, 1.9) | - | - | 2.1 | (1.0, 4.2) | 2.0 | (1.0, 4.1) |
| | Severe | 1.0 | (.6, 1.8) | - | - | 2.0 | (1.0, 4.0) | 2.1 | (1.1, 4.2) |
| Relationship dependencies | | | | | | | | | |
| | Drug financial dependency | 1.5 | (1.0, 2.4) | 1.3 | (.9, 2.0) | 1.9 | (1.1, 3.2) | 1.9 | (1.1, 3.2) |
| | Drug financial provider | 2.2 | (1.5, 3.4) | 1.9 | (1.2, 2.9) | 2.2 | (1.3, 3.7) | 2.2 | (1.3, 3.8) |
| | Pool money to buy drugs | 3.6 | (2.6, 5.0) | 2.9 | (2.0, 4.2) | 2.9 | (1.7, 4.9 | 2.8 | (1.6, 4.8) |
| | Do drugs or split drugs | 2.5 | (1.7, 3.7) | 2.2 | (1.5, 3.3) | 2.2 | (1.3, 3.8) | 2.1 | (1.2, 3.7) |
| | Scale | 1.4 | (1.2, 1.5) | 1.3 | (1.1, 1.5) | 1.4 | (1.2, 1.6) | 1.3 | (1.1, 1.6) |
| Childhood adversity | | 1.1 | (1.0, 1.3) | 1.1 | (1.0, 1.3) | 1.2 | (1.1, 1.4) | 1.2 | (1.1, 1.4) |
| Lifetime adversity | | 1.2 | (1.1, 1.3) | 1.2 | (1.0, 1.3) | 1.2 | (1.1, 1.3) | 1.2 | (1.1, 1.3) |

Adjusted models included covariates of race, ethnicity, education, marital status; RR: Relative Risk

with a 42% increase in risk of prior hospitalization due to substance use (RR$_{adjusted}$ = 1.4, 95% CI = 1.2, 1.7).

Participants who paid for their partners' drugs (RR$_{adjusted}$ = 1.5, 95% CI = 1.0, 2.4), pooled money to buy drugs (RR$_{adjusted}$ = 2.7, 95% CI = 1.7, 4.2), and did drugs or split drugs with their partners (RR$_{adjusted}$ = 1.9, 95% CI = 1.2, 3.0) were more likely to report ever witnessing an overdose. Each additional relationship dependency was associated with greater risk of ever witnessing an overdose (RR$_{adjusted}$ = 1.2, 95% CI = 1.08, 1.43). Participants who were dependent on their partners to purchase drugs (RR$_{adjusted}$ = 1.9, 95% CI = 1.1, 3.2), paid for their partners' drugs (RR$_{adjusted}$ = 2.2, 95% CI = 1.3, 3.8), pooled money to buy drugs (RR$_{adjusted}$ = 2.8, 95% CI = 1.6, 4.8), and did or split drugs with their partners (RR$_{adjusted}$ = 2.1, 95% CI = 1.2, 3.7) were more likely to report having a friend who experienced an overdose. Each additional relationship dependency was associated with an increase in the relative risk of having a family member who overdosed by 30% (RR$_{adjusted}$ = 1.3, 95% CI = 1.1, 1.6). Participants who paid for their partners' drugs (RR$_{adjusted}$ = 1.9, 95% CI = 1.2, 2.9), pooled money to buy drugs (RR$_{adjusted}$ = 2.9, 95% CI = 2.0, 4.2), and did or split drugs with their partners (RR$_{adjusted}$ = 2.2, 95% CI = 1.5, 3.3) were more likely to report having a friend who experienced an overdose. Each additional relationship dependency was associated with an increase in the risk of having a friend who experienced an overdose by 30% (RR$_{adjusted}$ = 1.3, 95% CI = 1.1, 1.5).

## Discussion

This study examined the association between several social factors of the risk environment including 1) exposures to 4 levels of IPV (none, verbal, moderate, severe), 2) drug-related relationship dependencies, and 3) history of childhood adversities and relative risk of experiencing an overdose, hospitalization because of drug use, witnessing an overdose and having friends or family who experienced an overdose among women who use drugs in New York City. Exposure to IPV was not associated with experiencing overdose. Findings from this paper supported hypotheses that exposure to moderate and severe IPV would be associated with ever

being hospitalized for drug use. Hospitalization due to losing consciousness, injury or drug poisoning may provide a marker for experiencing non-fatal overdose. In addition to IPV, women who paid for their partners' drugs, were dependent on their partners for money to purchase drugs, split drugs with their partners and pooled money to buy drugs were more likely to experience overdose, be hospitalized for drug use, witness an overdose, and have friends and family who had overdosed. Findings supported hypotheses that each cumulative exposure to childhood adversity was associated with increased risk of experiencing overdose, and ever being hospitalized for drug use. Future research must include multiple markers of overdose and overdose risk in assessments of overdose and drug use history.

Findings from this study are consistent with prior literature suggesting that exposures to childhood adversities are associated with increased risk of overdose and substance use severity among people who use illicit drugs.[27][48] Lake et al[27] found that exposures to childhood adversity was associated with increased risk of non-fatal overdose among people who inject drugs. To our knowledge, this is the first study to examine the association between IPV, relationship dependencies and overdose among women who use illicit drugs. However, findings from this study are congruent with previous literature that suggests exposure to IPV is significantly associated with substance use severity among women who use drugs.[16][49] Additional research is needed that further elucidates the relationships between IPV and overdose among women who use drugs.

## Implications for substance misuse treatment and overdose prevention

This paper generated several implications for substance use treatment and overdose prevention interventions as well as services for women exposed to intimate partner violence, drug-related relationship dependencies and childhood and lifetime adverse experiences. Research is needed that investigates if exposure to IPV may increase attitudinal barriers to treatment resulting in greater vulnerabilities to overdose. Future research must investigate whether women who have experienced IPV have lower self-efficacy, fewer positive attitudes toward treatment, and feelings of hopelessness creating barriers to accessing treatment. In addition to IPV, prior research suggests that exposure to childhood adversity is associated with lower health seeking behaviors among women who use drugs.[49]

Moreover, prior research suggests that trauma-focused interventions are needed in substance use treatment that acknowledge the vulnerabilities to substance use and relapse for people exposed to IPV and childhood adversity.[50] However thus far, IPV and childhood adversity have not been thoroughly integrated into overdose prevention interventions for women who use drugs despite prior literature suggesting that adversity-exposed women use drugs in greater quantities and higher frequencies.[15][16][22] Future studies must investigate whether providing interventions to reduce intimate partner violence may attenuate rapidly rising rates of overdose among women who use drugs.

Findings from this study are consistent with prior literature emphasizing the importance of assessing for the co-occurrence of prior IPV exposure and substance use in emergency departments and shelters for women.[11][51][52] Moreover, overdose risk assessments must consider exposure to IPV, childhood adversity and substance use in the distribution of naloxone and identifying individuals who could benefit from overdose education. Exposure to IPV and other forms of adversity may inhibit access to naloxone, naloxone training programs and education about overdose from non-opiates and other important overdose prevention resources. Future research must evaluate the potential benefits of distributing naloxone within shelters, case management and family justice centers on reducing the risk of overdose among women who experience IPV. To our knowledge there are no studies to date that evaluate interventions

to integrate harm reduction interventions into responses for women who experience intimate partner violence.

In addition to IPV and childhood adversity, mechanisms that link inequities in drug-related relationship dependencies and overdose have not been studied in prior literature. Women who depend on their intimate partners to pay for substances may be at greater risk of overdose when partners withhold payment for drugs or use financial support as a means of coercion. Prior literature suggests that financial and drug dependencies on intimate partners increases engagement in sex trading among women who use drugs who are in committed intimate relationships.[53] No prior studies have investigated the relationship between relationship dependencies and overdose among women who use drugs. Future research must examine whether financial dependency on partners to pay for drugs increases risk of overdose through mechanisms of lowered tolerance to substances, inconsistent quality of substances and increased engagement in overdose risk behaviors including injecting drugs. Additional research is needed that examines if women who depend on partners to procure their drugs may not know the quality of the substances resulting in greater risk of overdose. Studies must investigate whether women who rely on partners to provide drugs or split drugs are not aware of the quality of substances and consume doses that are too large, heightening their risk of overdose. Findings from this study suggests that additional research is necessary that investigates the benefits of providing couple-focused overdose prevention to drug-using couples that specifically addresses inequities in drug-related dependencies as a major risk factor for overdose in women.

## Limitations

There are several limitations of this study worth noting. The analysis used cross-sectional data thus precluding any causal inference, rather all statistical associations are associational. Participants were recruited by their male intimate partners who were in community corrections, thus generating a non-random sample selection limiting generalizability to the general population of women. Women who have died from overdose were not included in the sample, limiting generalization of findings to women who experienced non-fatal overdose. A number of variables were measured as lifetime experiences including experiencing overdose and hospitalizations due to drug use, witnessing an overdose and having friends and family who have experienced overdose. The cross-sectional design of the study and measurement of lifetime overdose tempers the strength of the findings in this manuscript. Overdose due to substance use may increase risk of experiencing intimate partner violence due to drug and financial relationship dependencies. However, our findings are consistent with prior longitudinal research that found exposure to partner violence at baseline increased engagement substance use.[54]

The small sample size used in this study precluded inclusion of other confounders that may potentially explain the association between IPV and overdose. The small number of overdoses that were reported by this sample also restricted the number of variables that could be included. This study did not include types of substances and injection drug use, in the multivariable regressions because too few participants overdosed across the types of substances. Nonetheless, these are important variables to consider in future analyses with larger sample sizes to take into account the role of substance use and other factors into the analyses of factors that may drive overdose among women who use illicit drugs. It is possible that underreporting occurred given that the research interviews occurred at probation and may have resulted in participants not disclosing overdose and drug use history.

Finally, male partner substance use behaviors were not included in this study and provide a fruitful avenue of future research. Prior studies suggest that male partner substance use is

associated with IPV severity which may exacerbate risk of overdose.[24][55][56] Future research must investigate how male-partner substance use interacts with male-partner perpetrated IPV to shape overdose risk environments of women who use drugs.

Nonetheless, findings from this study identify a population of women who are exposed to intimate partner violence at heightened risk of overdose. Future research is needed within a more recent timeframe to allow for correlational inference about the relationship between adversity and recent overdose experiences. Another limitation is this study did not investigate the potential underlying mechanisms that may be explaining associations between IPV and risk of overdose. Findings from this study call for future longitudinal epidemiological research to study the emergence of substance use and overdose following repeated exposures to adverse events. Due to issues of collinearity and sample size, all of the models examined risk factors separately that adjusted for several potential confounders. Future research with large sample sizes must examine how exposures to childhood adverse events interact with IPV to heighten overdose risk among women who use drugs.

## Conclusion

Limitations notwithstanding, identifying groups at greatest risk of overdose for prevention interventions including naloxone distribution, testing drug samples, and education could inform future research into strategies to attenuate rising rates of drug overdose among women. This paper addressed a significant gap in existing literature by elucidating several relationships between experience IPV, history of childhood and other lifetime adversity and several overdose-related factors including, experience of an overdose, hospitalization because of drug use, witnessing an overdose and having friends and family members who have experienced overdose. Future research must address the unique needs of women by expanding the incorporation of IPV, drug-related relationship inequities and childhood adversity into overdose prevention research.

## Supporting information

**S1 Dataset.**
(DTA)

## Author Contributions

**Conceptualization:** Nabila El-Bassel, Phillip L. Marotta, Tim Hunt, Louisa Gilbert.

**Data curation:** Dawn Goddard-Eckrich, Mingway Chang, Tim Hunt, Ewin Wu, Louisa Gilbert.

**Formal analysis:** Ewin Wu, Louisa Gilbert.

**Funding acquisition:** Nabila El-Bassel, Dawn Goddard-Eckrich, Tim Hunt, Ewin Wu, Louisa Gilbert.

**Investigation:** Nabila El-Bassel, Phillip L. Marotta.

**Methodology:** Phillip L. Marotta, Louisa Gilbert.

**Project administration:** Nabila El-Bassel, Dawn Goddard-Eckrich, Louisa Gilbert.

**Supervision:** Ewin Wu.

**Writing – original draft:** Nabila El-Bassel, Phillip L. Marotta, Louisa Gilbert.

**Writing – review & editing:** Nabila El-Bassel, Phillip L. Marotta, Louisa Gilbert.

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
