## [Decision Letter · Decision Letter 0]

12 Sep 2019

PONE-D-19-22408

Drug overdose among women in intimate relationships: The role of partner violence, trauma and relationship dependencies

PLOS ONE

Dear Dr. Marotta,

Thank you for submitting your manuscript to PLOS ONE. After careful consideration, we feel that it has merit but does not fully meet PLOS ONE’s publication criteria as it currently stands. Therefore, we invite you to submit a revised version of the manuscript that addresses the points raised during the review process.

As indicated by the Reviewers, required changes to the manuscript include addressing concerns about reverse causality, adjusting for known confounders, and developing the Discussion section for a more cohesive interpretation of the findings. 

We would appreciate receiving your revised manuscript by 10/11/19. To enhance the reproducibility of your results, we recommend that if applicable you deposit your laboratory protocols in protocols.io, where a protocol can be assigned its own identifier (DOI) such that it can be cited independently in the future. For instructions see: http://journals.plos.org/plosone/s/submission-guidelines#loc-laboratory-protocols

We look forward to receiving your revised manuscript.

Kind regards,

Javier Cepeda

Academic Editor

PLOS ONE

Journal Requirements:

3. We note you have included tables to which you do not refer in the text of your manuscript. Please ensure that you refer to Tables 1 - 4 in your text; if accepted, production will need this reference to link the reader to these Tables.

Reviewers' comments:

Reviewer's Responses to Questions

**Comments to the Author**

1. Is the manuscript technically sound, and do the data support the conclusions?

Reviewer #1: Yes

Reviewer #2: Yes

2. Has the statistical analysis been performed appropriately and rigorously? 

Reviewer #1: Yes

Reviewer #2: Yes

3. Have the authors made all data underlying the findings in their manuscript fully available?

Reviewer #1: No

Reviewer #2: No

4. Is the manuscript presented in an intelligible fashion and written in standard English?

Reviewer #1: Yes

Reviewer #2: Yes

5. Review Comments to the Author

Reviewer #1: This paper seeks to understand the relationship between IPV, experiences of traumatic events, relationship dependency and markers of drug overdose. Overall, it’s a very clear and concise paper, there are just a series of relatively minor issues I have with it, that need to be attended to.

First, I think the coding for IPV needs to be described in greater detail, and some of the terminology needs to be revised. In the abstract, IPV is wrongly described as a three level variables. Can the authors clarify their coding for minor versus severe phys/sex IPV experience please. In addition, could the authors reframe their language away from minor to maybe moderate IPV? I think the politics of describing violence against women as minor is worrying, given we know even small amounts of violence have negative health impacts. So this needs to be corrected throughout the paper.

P-values versus confidence intervals. Throughout the paper they have some pretty clear statistical language and are testing the null hypothesis, yet at times the authors then use p>0.05 to show association. Looking at the confidence intervals, it seems it’s because of it including 1. Given the small sample and the push away from p-values, I wonder if it would be better to drop p-values and just used 95%CI for the tables and paper. If the authors don’t then they need to revise the paper to be strict on the p<0.05 statistical rules etc.

In terms of language, I wonder if it’s right to describe witnessing violence etc. as trauma, or traumatic events, as trauma would be the psychological outcomes, e.g. PTSD.

In the discussion section, there is no previous literature cited – is this truly the case? The only time literature comes in is around implications. I would have imagined there would be a little bit of previous work on this etc.

Minor issues

In the childhood trauma measure section, the first question about verbal trauma, was this before the age 17 or ever? If ever, I wonder if this needs dropping, as it’s quite different to the other questions – or did it just get missed out in the typing.

In the description of the statistical analyses, they say they are going to give standard error estimates – surely inter-quartile range looking at the tables?

In descriptive stats section, “2.00 for lifetime traumas (0-6)” just a few more words needed to make this interpretable.

In the discussion the authors state:”A greater number of women reported experiencing hospitalization from drug use suggesting that hospitalization from drug use may provide a more accurate marker of self-reported non-fatal overdose than asking about overdose.” I think this needs unpacking further, as it’s not clearly established by the analyses presented here.

I could not see clear evidence that the paper included access to the data set as per PLoS guidelines.

Other than this I think the paper is very well written.

Reviewer #2: This study examines the trauma- and drug-related correlates of five outcomes (overdose, hospitalization for drug use, and witnessing overdose) using a convenience sample of partnered women recruited through a randomized HIV prevention intervention.

Though the authors have identified a neglected area of research, there were notable methodological and presentation issues with the manuscript. The use of lifetime outcomes (rather than recent events) tempered my enthusiasm for this exploratory analysis as was difficult to rule out reverse causality. Key confounders were missing in the models (e.g., age, types of drugs used, injection drug use, incarceration history). Given that the eligibility criteria for drug use was broad, the analysis likely includes a heterogeneous sample of drug-using women with varying levels of overdose-related risk, which was a concern.

The introduction was unusually long and did not cohesively frame the research hypotheses. For example, the exploration of hospitalization for drug use was not justified in the risk environment conceptual framework used. In fact, the authors cited multiple papers conceptualizing the risk environment – given the evolution of this framework in the fields of HIV and opioid research, better articulation will be needed to guide the reader. Conversely, the discussion was surprisingly underdeveloped and not grounded in the aforementioned literature. This manuscript needs significant revision.

Minor comment:

There were several typographical errors.

6. PLOS authors have the option to publish the peer review history of their article (what does this mean?). If published, this will include your full peer review and any attached files.

Reviewer #1: Yes: Andrew Gibbs

Reviewer #2: Yes: Ju Nyeong Park

---

## [Author Response · Author response to Decision Letter 0]

28 Oct 2019

Dear Dr. Cepeda, 

We are delighted to provide the following revised manuscript for consideration in the special issue “Substance Use, Misuse and Dependence: Prevention and Treatment.” We greatly appreciate the feedback provided by you and the two peer reviewers. We agree with all of the feedback and comments raised by the reviewers. We revised the manuscript and addressed the comments raised by the reviewers. We provided a marked-up copy of our manuscript that highlights changes made to the original version and is uploaded as a separate file and labeled as ‘revised manuscript with tracked changes.’ We also included an unmarked version of our revised paper without tracked changes and is labeled ‘manuscript.’ To enhance the reproducibility and transparency of our results and to meet the requirements of PLOS one, we deposited the data underlying findings from this study in Columbia University academic commons. We will provide repository information with the DOI number for the dataset. The revised manuscript meets PLOS ONE’s style requirements. 

We provide a point-by-point response below to all of the feedback raised by the editor and peer reviewers.

Associate Editor

 “We note that you have stated that you will provide repository information for your data at acceptance. Should your manuscript be accepted for publication, we will hold it until you provide the relevant accession numbers or DOIs necessary to access your data. If you wish to make changes to your Data Availability statement, please describe these changes in your cover letter and we will update your Data Availability statement to reflect the information you provide.”

We are committed to complying with providing repository information for our data at acceptance of this article. 

There are no ethical or legal restrictions on sharing our de-identified data set. We deposited our data in Columbia Academic Commons and will share a DOI with the available data, as well as contact information for the data access committee at Columbia University if the manuscript is accepted. We would be delighted to provide any additional information.

“We note you have included tables to which you do not refer in the text of your manuscript. Please ensure that you refer to Tables 1 - 4 in your text; if accepted, production will need this reference to link the reader to these Tables.”

The authors now refer to all the tables in the text of the manuscript (Tables 1-4) so that production will have this to link the readers to the tables. 

“Have the authors made all data underlying the findings in their manuscript fully available?”

We deposited our data in Columbia Academic Commons and will share a DOI with the available data, as well as contact information for the data access committee at Columbia University if the manuscript is accepted. We would be delighted to provide any additional information.

Reviewer 1:

“The coding for IPV needs to be described in greater detail, and some of the terminology needs to be revised. In the abstract, IPV is wrongly described as a three-level variables. Could the authors clarify their coding for minor versus severe phys./sex IPV experience please?”

We agree with the reviewers and clarified the description of coding for minor versus severe phys./sex IPV to accurately reflect what was done to the variable. The IPV variable was created as none (0), verbal only (1), moderate only (2) and major abuse (3). 

The following revisions to the methods section of the manuscript:

“A 4-level categorical variable was created with categories measuring exposure to: 0) no physical/sexual, 1) verbal aggression only (no moderate /severe physical or sexual abuse 2) only moderate and 3) severe sexual/physical abuse.”

“In addition, could the authors reframe their language away from minor to maybe moderate IPV? I think the politics of describing violence against women as minor is worrying, given we know even small amounts of violence have negative health impacts. So, this needs to be corrected throughout the paper.”

We revised the manuscript throughout to replace the use of “minor” with “moderate.” The revised manuscript reflects these changes. 

“P-values versus confidence intervals. Throughout the paper they have some pretty clear statistical language and are testing the null hypothesis, yet at times the authors then use p>0.05 to show association. Looking at the confidence intervals, it seems it’s because of it including 1. Given the small sample and the push away from p-values, I wonder if it would be better to drop p-values and just used 95%CI for the tables and paper. If the authors don’t then they need to revise the paper to be strict on the p<0.05 statistical rules etc.”

Initially we presented findings with p<.10 as the threshold for statistical significance. We agree with the push away from p-values. Based on recommendations from the reviewers we decided to only include confidence intervals. We deleted all reference to p-values and only use confidence intervals throughout. 

“In terms of language, I wonder if it’s right to describe witnessing violence etc. as trauma, or traumatic events, as trauma would be the psychological outcomes, e.g. PTSD.”

We agree with this feedback and revised the manuscript to include only use of “adverse events” or “adversity” to replace use of Trauma. 

“In the discussion section, there is no previous literature cited – is this truly the case? The only time literature comes in is around implications. I would have imagined there would be a little bit of previous work on this etc.”

We agree with this feedback and significantly restructured the discussion section to include more citations and we contextualize our research findings in existing research. To our knowledge there are no previously published studies that examine the relationship between intimate partner violence and overdose. There is however additional research on the relationship between exposure to childhood adverse events and overdose. In addition to childhood adversity, a robust body of literature exists connecting IPV to substance use severity. As suggested we endeavored to incorporate this into the discussion section.

In addition to changes throughout the discussion section we included the following: 

“Findings from this study are consistent with prior literature suggesting that exposures to childhood adversities are associated with increased risk of overdose and substance use severity among people who use illicit drugs (Lake et al., 2015; Sacks et al., 2008). Lake et al., (2015) found that exposures to childhood trauma was associated with increased risk of non-fatal overdose among people who inject drugs. To our knowledge ,this is the first study to examine the association between IPV, relationship dependencies and overdose among women who use illicit drugs. However, findings from this study are congruent with previous literature that suggests exposure to IPV is significantly associated with substance use severity among women who use drugs ; Salomon et al., 2002; Testa et al., 2003). Additional research is needed that further elucidates the relationships between IPV and overdose among women who use drugs.” 

We also included several additional citations to address this suggestion. 

Minor issues

“In the childhood trauma measure section, the first question about verbal trauma, was this before the age 17 or ever? If ever, I wonder if this needs dropping, as it’s quite different to the other questions – or did it just get missed out in the typing.”

The childhood trauma measure (now referenced as adversity) asked about verbal trauma before the age of 17. The authors added “before the age of 18” to accurately reflect how the question was worded

“In the description of the statistical analyses, they say they are going to give standard error estimates – surely inter-quartile range looking at the tables?”

Yes, we provided IQR for the descriptive tables. The manuscript now reflects the provision of this descriptive statistic (IQR)

“In descriptive stats section, “2.00 for lifetime traumas (0-6)” just a few more words needed to make this interpretable.”

The following was included to make this interpretable: 

“The mean score for relationship dependencies was 2.00 experiences (0-6).”

“In the discussion the authors state:” A greater number of women reported experiencing hospitalization from drug use suggesting that hospitalization from drug use may provide a more accurate marker of self-reported non-fatal overdose than asking about overdose.” I think this needs unpacking further, as it’s not clearly established by the analyses presented here.”

The authors decided to omit this sentence from the manuscript

“I could not see clear evidence that the paper included access to the data set as per PLoS guidelines.”

We deposited this dataset in academic commons at Columbia University and will provide the DOI upon acceptance of this manuscript. 

“Other than this I think the paper is very well written.”

Thank you 

Reviewer #2:

“The use of lifetime outcomes (rather than recent events) tempered my enthusiasm for this exploratory analysis as was difficult to rule out reverse causality. 

We agree that lifetime outcomes and cross-sectional data temper the strength of the findings in this manuscript. We believe that our exploratory analysis presents a compelling case for future research. Nonetheless, we emphasize this limitation in the revised manuscript. We included the following in the manuscript: 

“The cross-sectional design of the study and measurement of lifetime overdose tempers the strength of the findings in this manuscript. Overdose due to substance use may increase risk of experiencing intimate partner violence due to drug and financial relationship dependencies. However, our findings are consistent with prior longitudinal research that found exposure to partner violence at baseline increased engagement substance use (Salomon et al., 2002). 

“Key confounders were missing in the models (e.g., age, types of drugs used, injection drug use, incarceration history). Given that the eligibility criteria for drug use was broad, the analysis likely includes a heterogeneous sample of drug-using women with varying levels of overdose-related risk, which was a concern.”

We agree with the importance of confounders that were not included in the regression models. The sample included a heterogeneous sample of women who use drugs with varying levels of overdose-related risk. We were limited by the small sample size and thus were conservative with the number of covariates included in the models. We have noted this limitation in the discussion section. For several of the variables the responses were too low to include in the models including types of drugs used and injection drug use. We included this information descriptively for the readers’ benefit. To address these concerns we conducted a sensitivity analysis by including age in the multivariable model which was insignificantly associated with all of the Overdose outcomes and did not significantly change the results of the associations between IPV, childhood adversity and overdose outcomes. We included the following in the limitation section:

The following was added to the methods section:

“We performed sensitivity analysis with other variables with age, binge drinking and several types substances. None of these variables were significant and due to the small sample size were not included in the final regression model estimating where we examine the association between IPV and overdose.”

The following was added to the discussion section:

“The small sample size used in this study precluded inclusion of other confounders that may potentially explain the association between IPV and overdose. The small number of overdoses that were reported by this sample also restricted the number of variables that could be included. This study did not include types of substances and injection drug use, in the multivariable regressions because too few participants overdosed across the types of substances. Nonetheless, these are important variables to consider in future analyses with larger sample sizes to take into account the role of substance use and other factors into the analyses of factors that may drive overdose among women who use illicit drugs. Additionally, it is possible that underreporting occurred given that the research interviews occurred at probation and may have resulted in participants not disclosing overdose and drug use history.”

 “The introduction was unusually long and did not cohesively frame the research hypotheses. For example, the exploration of hospitalization for drug use was not justified in the risk environment conceptual framework used. In fact, the authors cited multiple papers conceptualizing the risk environment – given the evolution of this framework in the fields of HIV and opioid research, better articulation will be needed to guide the reader.”

We shortened the introduction. We also restructured the introduction throughout to better frame the research hypotheses. We agree that hospitalization as a dimension of the risk environment needed more development. To address these concerns, we added the following: 

“Prior research has shown that following hospitalization for drug use, people who use opioids are at greater risk of returning to drug use and overdosing due to lower tolerance and experiencing symptoms of withdrawal (Papp & Schrock, 2017). Emergency department hospitalization is an important yet understudied feature of the overdose risk environment for women who use opioids. Hospitals may be an opportune setting to deliver both overdose prevention through naloxone distribution and IPV services for women who are hospitalized due to drug use (Papp & Schrock, 2017). No studies to date have examined the associations among women between IPV and the specific outcomes of overdose, hospitalization from drug use, witnessing overdose, and having friends or family who have overdosed.”

“Conversely, the discussion was surprisingly underdeveloped and not grounded in the aforementioned literature.”

We agree with this feedback and significantly developed the discussion section. We added more citations from the aforementioned literature as well as additional text contextualizing how our research findings fit with extant literature on IPV and substance use among women. Specifically ,we provide citations and note where our findings are consistent with prior literature. The conclusion is now grounded in the aforementioned literature. 

Minor comment:

There were several typographical errors.

The authors extensively edited this manuscript for typographical errors.

---

## [Editor Report · Decision Letter 1]

14 Nov 2019

Drug overdose among women in intimate relationships: The role of partner violence, adversity and relationship dependencies

PONE-D-19-22408R1

Dear Dr. Marotta,

We are pleased to inform you that your manuscript has been judged scientifically suitable for publication and will be formally accepted for publication once it complies with all outstanding technical requirements.

With kind regards,

Giuseppe Carrà, PhD

Academic Editor

PLOS ONE
---

## [Editor Report · Acceptance letter]

3 Dec 2019

PONE-D-19-22408R1 

Drug overdose among women in intimate relationships: The role of partner violence, adversity and relationship dependencies 

Dear Dr. Marotta:

I am pleased to inform you that your manuscript has been deemed suitable for publication in PLOS ONE. Congratulations! Your manuscript is now with our production department. 

With kind regards,

on behalf of

Dr. Giuseppe Carrà 

Academic Editor

PLOS ONE